# Lemur: Harmonizing Natural Language and Code for Language Agents

**Yiheng Xu**[*♠◇] **Hongjin Su**[*♠◇] **Chen Xing**[*♠] **Boyu Mi**[♠◇] **Qian Liu**[♡◇] **Weijia Shi**[△]
**Binyuan Hui**[◇] **Fan Zhou**[♠◇] **Yitao Liu**[♠◇] **Tianbao Xie**[♠◇] **Zhoujun Cheng**[♠◇] **Siheng Zhao**[♠◇]
**Lingpeng Kong**[♠] **Bailin Wang**[★] **Caiming Xiong**[♠] **Tao Yu**[♠◇]
[♠]University of Hong Kong   [◇]XLang Lab   [♠]Salesforce Research
[♡]Sea AI Lab   [△]University of Washington   [★]MIT CSAIL
[♠]{yhxu,hjsu,tyu}@cs.hku.hk   [♠]{cxing, cxiong}@salesforce.com

## Abstract

We introduce Lemur and Lemur-Chat, openly accessible language models optimized for both natural language and coding capabilities to serve as the backbone of versatile language agents. The evolution from language chat models to functional language agents demands that models not only master human interaction, reasoning, and planning but also ensure grounding in the relevant environments. This calls for a harmonious blend of language and coding capabilities in the models. Lemur and Lemur-Chat are proposed to address this necessity, demonstrating balanced proficiencies in both domains, unlike existing open-source models that tend to specialize in either. Through meticulous pretraining using a code-intensive corpus and instruction fine-tuning on text and code data, our models achieve state-of-the-art averaged performance across diverse text and coding benchmarks. Comprehensive experiments demonstrate Lemur's superiority over existing open-source models and its proficiency across various agent tasks involving human communication, tool usage, and interaction under fully- and partially- observable environments. The harmonization between natural and programming languages enables Lemur-Chat to significantly narrow the gap with proprietary models on agent abilities, providing key insights into developing advanced open-source agents adept at reasoning, planning, and operating seamlessly across environments. Our model and code have been open-sourced at https://github.com/OpenLemur/Lemur.

## 1 Introduction

Intelligent agents are broadly conceptualized as autonomous problem solvers with the abilities to sense their environment, decide, and act upon that enviorment (Sutton & Barto, 2005; Russell, 2010; Wilkins, 2014). Recent implementations of this concept in creating language agents (Yao et al., 2022b; Gravitas, 2023; Wang et al., 2023a) capable of utilizing natural language for varied and intricate tasks in diverse environments have demonstrated potential, particularly when built upon large language models (LLMs) (Brown et al., 2020; Chen et al., 2021; Chowdhery et al., 2022; OpenAI, 2023; Touvron et al., 2023a). Such agents harness the human knowledge in LLMs and can think and communicate in human terms. This equips them to employ varied tools, operate in complex environments, engage in language reasoning, and create spontaneous multi-agent systems.

To effectively form the foundation of language agents, LLMs should not only master human interaction, reasoning, and planning but also ensure grounding in the relevant environments (Wei et al., 2022; Huang et al., 2022a; Ichter et al., 2022). Human interaction, reasoning, and planning can be largely realized through the natural language capabilities of LLMs. On the other hand, the grounded execution in the environment is usually achieved by using general-purpose code or domain-specific APIs, such as controlling web browsers (Shi et al., 2017; Yao et al., 2022a; Deng et al., 2023; Zhou

---

[*]Equal contribution.

et al., 2023), interacting with OS CLI terminals (Yang et al., 2023), and manipulating robotic arms via primitive APIs (Ichter et al., 2022; Huang et al., 2022b; Liang et al., 2023). Therefore, we posit that for the construction of language agents, it is imperative for language models to possess harmonized capabilities in both natural language and programming languages. This balance ensures that models do not specialize exclusively in certain areas but can seamlessly integrate with environment contexts and generate controllable and valid actions. Presently, closed-source models like GPT-4 (OpenAI, 2023) demonstrate such capabilities, which empower them to function as language agents. However, current open-source LLMs such as Llama 2 (Touvron et al., 2023a) and CodeLlama (Rozière et al., 2023) have traditionally been tailored for either textual or code-related tasks, with limited ability to effectively balance both.

To address this need, we introduce Lemur and Lemur-Chat, cutting-edge, openly accessible models pre-trained and fine-tuned to harmonize text and code capabilities. We enhanced the base Llama-2-70B through thoughtfully designed pre-training and instruction fine-tuning stages. Specifically, we built a code-centric corpus based on The Stack (Kocetkov et al., 2022), comprising 90B tokens with a 10:1 code-to-text ratio, ensuring improved capabilities in coding ability while maintaining performance in natural language ability. We refer to this model as Lemur. After pretraining, we conducted instruction fine-tuning using about 300K examples from both text and code to build an instruction-following model, which we refer to as Lemur-Chat. Thorough assessments across 8 textual and coding benchmarks validate the superior performance of both Lemur and Lemur-Chat in multiple text and code evaluations, establishing them as the most well-rounded open-source models.

Moreover, this work embarks on assessing the vital capabilities of language agents across various scenarios, which we refer to as agent benchmarks. We place a particular emphasis on their tool-usage abilities, and abilities in grounding in the environment feedback and human feedback. We also explore the challenges posed by real and partially observable environments where the agent has to take actions based on limited knowledge and take additional actions to gather more information. Experimental results indicate that Lemur-Chat outperforms other open-sourced models in 12 of the 13 agent benchmarks. This underscores how the integration of natural and coding capabilities allows Lemur-Chat to exceed the current open-source models for language agents, markedly bridging the performance disparity between open-source and commercial alternatives. Our experiments show an important insight that in agent scenarios, there is a need for synergy between natural language and coding abilities. Specifically, for models with strong natural language capabilities but weak coding abilities like `Llama-2-70B-Chat`, they can effectively use simple tools to assist reasoning (§4.2) because the action space is small and the difficulty of using tools is low. However, when facing complex decision-making scenarios such as web browsing and house navigation, the action space is usually large, and models with strong coding abilities have an advantage in generating complex executable action sequences (§4.5). Moreover, we did error analysis on several environments covering multiple agent skills to reflect the key challenges for future language agent development, including tool and action executability §D.1, problem difficulty §D.2, and domain knowledge §D.3. Overall, Lemur has both strong natural language and coding abilities, enabling it to achieve better performance in both scenarios. This research provides insights into optimizing the synergy between natural and programming languages, laying a solid foundation for developing advanced language agents capable of operating efficiently in various environments.

## 2    PRE-TRAINING AND INSTRUCTION TUNING OF LEMUR

This section will introduce the method to build the Lemur and Lemur-Chat models and their performance on commonly-used benchmarks for pre-trained language model evaluation. To build a more balanced model, the pipeline includes two stages: pre-training (§ 2.1) and instruction fine-tuning (§ 2.2). The training pipeline is shown in Figure 1.

### 2.1    PRE-TRAINING

In the pre-training stage, we choose the `Llama-2-70B` base model as the starting point, which is the cutting-edge open-sourced base model in most scenarios except the coding scenario. Our goal is to improve the coding ability while maintaining the reasoning ability of Llama-2. To this end, we built a corpus with a code-to-text ratio of 10:1. We discuss how we decided this ratio in § A.1.2. For the code part, we base it on The Stack (Kocetkov et al., 2022), a collection of source

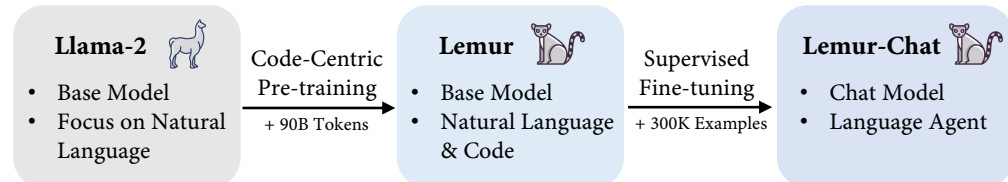

Figure 1: Overview of Training Procedure. We continually pre-train `Llama-2-70B` model on 90B code-intensive data and fine-tune it with 300K examples of instructions to enable the model to harmonize natural language and coding abilities.

Table 1: Instruction datasets investigated in this work. We report the average number of rounds ($\bar{N}_{\text{rounds}}$), average length of prompts ($\bar{L}_{\text{prompt}}$), average length of completion ($\bar{L}_{\text{completion}}$).

| Datasets | Query Source | Response Source | # Instances | $\bar{N}_{\text{rounds}}$ | $\bar{L}_{\text{prompt}}$ | $\bar{L}_{\text{completion}}$ |
|---|---|---|---|---|---|---|
| Open Assistant 1 | Human-written | Human-written | 34,546 | 1.6 | 34.9 | 213.1 |
| OpenOrca | Human-written | GPT-4 | 200,000 | 1.0 | 242.8 | 174.1 |
| ShareGPT & ChatLogs | User prompts | GPT-3.5/GPT-4 | 81,319 | 6.0 | 96.7 | 340.2 |
| Evol-CodeAlpaca | GPT-3.5/GPT-4 | GPT-3.5/GPT-4 | 51,952 | 1.0 | 93.9 | 343.5 |

codes from GitHub with permissive licenses. Among all languages, we focus on scripting or interpreted languages (Python, SQL, Bash, Perl, etc.) because agent models are often executed in interactive scenarios. Unlike compiled languages like C++, the interpreted nature of scripting languages allows for immediate execution and ease of modification, which is essential for dynamic interaction in language agent scenarios. As for the text aspect, we use RefinedWeb (Penedo et al., 2023), Redpajama (Computer, 2023), as well as CommonCrawl, Wikipedia, Books, ArXiv, Stack-Exchange and DM Mathematics (Saxton et al., 2019) to build the textual data corpus. Following previous works (Gao et al., 2020; Smith et al., 2022; Computer, 2023; Li et al., 2023), we do extensive deduplication after aggregating all data sources. The composition of the data is shown in Appendix § A.1.1. We train the `Lemur-70B` model initialized with `Llama-2-70B` using a TPUv4-512 pod. We train the model with sequence packing (Raffel et al., 2019; Chung et al., 2022) to improve the training efficiency. Please refer to Appendix § A.1.3 for more details.

## 2.2 INSTRUCTION FINE-TUNING

During the instruction fine-tuning phase, we include four data sources to construct Lemur-Chat, including the Open Assistant crowdsourced annotated dialogue corpus (Köpf et al., 2023), Orca data with chain of thought reasoning for human-written tasks (Lian et al., 2023; Mukherjee et al., 2023), ShareGPT & Chatlogs containing real user and ChatGPT history records (ShareGPT data), as well as Evol-CodeAlpaca data (Luo et al., 2023) consisting of complex coding tasks generated by ChatGPT along with their solutions. The statistics of these instruction datasets are shown in Table 1. After we collect these data, we additionally clean and deduplicate these instruction fine-tuning data. We conduct training on these data for 2 epochs. Please refer to § A.2 for more details.

## 3 FROM LANGUAGE MODEL TO LANGUAGE AGENT

This section introduces how we measure the language and coding abilities of a language model to guide the process of harmonization. We further discuss the new challenges faced when connecting LLM to the environment and describe how we examine the necessary agent capabilities.

### 3.1 FUNDAMENTAL LANGAUGE AND CODING CAPABILITIES

Previous work typically uses a variety of benchmarks to comprehensively reflect the performance of models on different types of tasks. Therefore, we evaluate the performance of various models across text and code benchmarks as follows.

- *Text benchmarks:* MMLU (Hendrycks et al., 2021a) to determine factuality, BBH (Suzgun et al., 2022) to check reasoning abilities, GSM8K (Cobbe et al., 2021) to gauge math reasoning.
- *Code benchmarks:* HumanEval (Chen et al., 2021) and MBPP (Austin et al., 2021) to test Python writing abilities, Spider (Yu et al., 2018) to assess database query capabilities, MultiPL-E (Cassano et al., 2023) to measure the multi-lingual coding capabilities, DS-1000 (Lai et al., 2023) for evaluation in data science scenarios

## 3.2 CONNECTING LLM AGENTS TO ENVIRONMENT

While the measurements in §3.1 provide valuable insights on each domain, they may not fully encapsulate the models' capabilities in language agent scenarios due to their focus on single-turn interactions in fully observable settings. In order to address this discrepancy, we scrutinize models in the context of multi-turn interactive scenarios to gauge their adaptability in practical situations. Our assessment is centered on various capabilities of language agents, which are encapsulated in the factors outlined in Figure 2 and Table 2. For a more comprehensive assessment of these capabilities, we reorganize existing multiple datasets into four sets to examine the diverse skills demanded by the agent scenario. Please refer to Appendix B for details of each dataset.

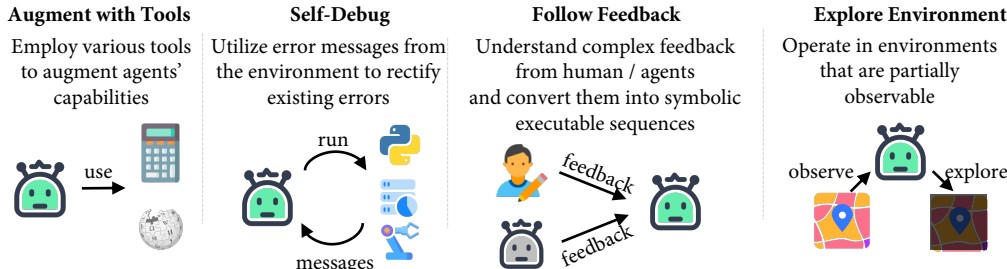

Figure 2: We inspect language agents in various aspects, including the abilities to augment with tools, self-debug, follow feedback, and explore partially observable environments.

Table 2: Multi-turn agent evaluation under different settings. We evaluate crucial capabilities for LLM as an agent, spanning tool usage, feedback adherence and environment exploration.

| Capabilities | Environments | Observability | Datasets |
|---|---|---|---|
| Augment with Tools (§4.2) | Python Calculator
WikiSearch | Fully | M-{GSM8K, MATH, TheoremQA}
M-{HotpotQA, MMLU} |
| Self-debug with
Environment Feedback (§4.3) | Python
OS Terminal
Database
Robotics API | Fully | M-{HumanEval, MBPP}
InterCode-Bash
InterCode-SQL
RoboCodeGen |
| Follow Natural Language
Feedback (§4.4) | User/Agent
User/Agent | Fully | M-Reasoning w/ GPT-4 feedback
M-Code w/ GPT-4 feedback |
| Explore in Partially
Observable Environment (§4.5) | OS Terminal
Web Browser
Embodied Simulator | Partially | InterCode-CTF
WebArena
M-ALFWorld |

**Augment with Tools** Tool usage (Schick et al., 2023; Hao et al., 2023; Mialon et al., 2023) is an important capability of language agents. Tasks like calculations or information retrieval can be offloaded to external modules (e.g. Python interpreters or search engines) using tools (Gao et al., 2023; Chen et al., 2022), improving reliability and interpretability. Tools involve calling operators in symbolic languages or invoking APIs (Shen et al., 2023). This requires decomposing a task, grounding intention to tools, and using the results for further actions, relying on natural language reasoning and programming abilities (Cheng et al., 2023; Surís et al., 2023). To assess the ability of language agents to solve complex multi-turn problems using tools, we introduce the part of the MINT

dataset (Wang et al., 2023b) which focuses on tool-utilization for reasoning. This part includes several adapted datasets for testing: MINT-{GSM8K, MATH} (Cobbe et al., 2021; Hendrycks et al., 2021b) is used to test the model's ability to solve mathematical problems using a Python interpreter, and MINT-{HotpotQA, MMLU} (Yang et al., 2018; Hendrycks et al., 2021a) assesses the model's capability to solve knowledge-based questions using Wikipedia searches. At the same time, for the MINT-TheoremQA (Chen et al., 2023a), the model needs to perform knowledge searches on Wikipedia and use a Python interpreter to calculate and draw conclusions.

**Self-debug with Environment Feedback** Self-debug is an important way to test whether a model can incorporate environmental feedback (Jignasu et al., 2023; Olausson et al., 2023; Chen et al., 2023b). In the self-debug scenario, the model usually needs to complete a complex operation sequence, such as Python functions, database queries/modifications, robot action sequences, etc (Gur et al., 2023; Yao et al., 2023). These complex operations sometimes cannot be executed successfully and will return error messages. This requires the model to comprehend this kind of environmental feedback and correct errors, which tests the joint effect of natural language reasoning ability and programming languages. We use rich datasets from multiple benchmarks to evaluate this performance, including multi-turn MINT-MBPP and MINT-HumanEval in MINT (Wang et al., 2023b), SQL and Bash in InterCode (Yang et al., 2023), as well as RoboCodeGen that calls robot APIs through code (Liang et al., 2023). These environments require the model to complete complex tasks and will provide execution errors. In these environments, model performance will vary according to its self-debugging ability, reflecting the ability to incorporate feedback.

**Follow Natural Language Feedback** Following natural language feedback is an important mechanism for agents to receive information from humans or other agents (Wang et al., 2022a; Ouyang et al., 2022; Gong et al., 2023). In scenarios where complex problems are solved through multi-turn interactions, the model not only needs to incorporate environmental feedback but also feedback from humans or other agents in order to improve. This mechanism requires the model to understand new instructions based on a context that combines natural language and code, and ground them into new action sequences. To evaluate the model's ability to accept natural language feedback, we follow the approach of the MINT benchmarks: using a GPT-4 simulated user as a teacher to guide the model in problem-solving. This setup includes a series of MINT datasets (Wang et al., 2023b) to comprehensively evaluate performance after adding natural language feedback in various scenarios.

**Explore in Partially Observable Environments** Exploring partially observable environments is a unique and challenging factor in agent scenarios. All the settings mentioned earlier can be considered as fully observable environments, which means that agents can observe all the information of the environment to plan, reason, and make decisions. However, in partially observable environments (also known as Partially Observable Markov Decision Process) (Kurniawati, 2021), agents can only partially observe the environmental information to solve problems. This requires agents to collect information through exploration and continue making decisions. This process places high demands on various abilities of agents, such as natural language planning and reasoning, environmental interaction, etc. To measure this ability, we use three datasets: InterCode-CTF (Yang et al., 2023) and WebArena (Zhou et al., 2023) in digital environments, as well as ALFWorld (Shridhar et al., 2020b) in physical environments. InterCode-CTF provides an OS terminal for models to solve Catch the Flag (CTF) problems where agents need multiple rounds of exploration to obtain the flag. WebArena evaluates agents' ability to control browsers for task completion through exploration. ALFWorld is a simulated home environment where agents need to explore navigation and complete specific tasks.

## 4 EXPERIMENTAL RESULTS

### 4.1 LANGUAGE AND CODE CAPABILITIES

The comprehensive evaluations of text and code benchmarks in Table 3 demonstrate the impressive capabilities of `Lemur-70B` and `Lemur-70B-Chat` models. Deviating from `Llama-2-70B` and `Llama-2-70B-Chat`, which are mostly pre-trained and fine-tuned on text data, the Lemur models augment coding abilities and thereby enhance the overall performance by 4.3% and 14.8% respectively. Alternatively, models like StarCoder-15B, WizardCoder-15B, `CodeLlama-34B` and `CodeLlama-34B-INST`, which are primarily trained on code datasets, exhibit solid performance in code benchmarks. On average, `Lemur-70B` outperforms StarCoder-15B, StarCoderPlus-15B and `CodeLlama-34B` by 14.3%, 16.8% and 1.9% respectively, and `Lemur-70B-Chat` surpasses

Table 3: Performance comparison across diverse models on text and code benchmarks. MCode is an abbreviation for Multilingual Code. HE stands for HumanEval. Avg denotes the average performance across all benchmarks. `Lemur-70B` and `Lemur-70B-Chat` exhibit balanced capabilities, achieving the highest overall performance when averaged by task. See Appendix C for more details.

| Model | Text | | | Code | | | | | Avg |
| | QA | Reason | Math | Python | | SQL | MCode | DS | |
| | MMLU | BBH | GSM8K | HE | MBPP | Spider | MultiPL-E | DS-1000 | |
|---|---|---|---|---|---|---|---|---|---|
| StarCoder-15B | 30.8 | 33.2 | 8.9 | 33.6 | 52.7 | 58.3 | 25.3 | 26.0 | 33.6 |
| StarCoderPlus-15B | 42.0 | 36.2 | 17.7 | 26.2 | 37.0 | 48.8 | 21.4 | 19.4 | 31.1 |
| CodeLlama-34B | 52.8 | 42.2 | 32.7 | 48.8 | 55.0 | 68.4 | 36.4 | 31.8 | 46.0 |
| Llama-2-70B | 68.9 | 51.2 | 56.8 | 30.5 | 45.4 | 60.0 | 24.4 | 11.3 | 43.6 |
| Lemur-70B | 64.5 | 51.6 | 54.9 | 35.4 | 53.2 | 62.8 | 30.4 | 30.7 | **47.9** |
| WizardCoder-15B | 29.4 | 28.8 | 7.1 | 57.3 | 51.6 | 61.6 | 30.8 | 29.2 | 37.0 |
| CodeLlama-34B-INST | 53.5 | 37.1 | 41.0 | 41.5 | 57.0 | 66.6 | 36.1 | 32.3 | 45.6 |
| Llama-2-70B-Chat | 63.9 | 38.9 | 48.7 | 31.1 | 38.2 | 60.3 | 22.6 | 17.8 | 40.2 |
| Lemur-70B-Chat | 65.3 | 61.9 | 66.3 | 61.0 | 55.5 | 62.5 | 42.9 | 34.5 | **55.0** |

WizardCoder-15B and `CodeLlama-34B-INST` by 18.0% and 9.4% respectively. This commendable increase highlights the virtue of harmonizing textual and coding skills.

The synergic text and code abilities enable them to function as language agents. However, a disparity exists between natural language and coding capabilities in current open-source models. Such limitations obstruct these models' abilities to act as language agents, leading to performance degradation in agent benchmarks. In subsequent sections, we meticulously evaluate various critical capabilities of agents, revealing the importance of synergic text and code abilities to language models.

## 4.2 AUGMENT WITH TOOLS

In the realm of problem-solving, agents, akin to humans, employ various tools to augment their capabilities. This is exemplified by (Chen et al., 2022), who showcase that the mathematical reasoning prowess of LLM can be significantly enhanced with the aid of Python. As per the data presented in Table 4, it is evident that `Lemur-70B-Chat` outperforms both `Llama-2-70B-Chat` and `CodeLlama-34B-INST`, indicating its superior ability to effectively leverage tools. In addition, we found performance differences between open-source models and closed-source models in the challenging math benchmark tests M-MATH and M-TheoremQA. We further analyzed the errors to better understand this phenomenon, please refer to the content in Appendix D.1.

Table 4: The tool-augmented reasoning tasks evaluate the model's capabilities to use tools in the reasoning process. Across five tasks with Python and WikiSearch API as tools, `Lemur-70B-Chat` outperforms both `Llama-2-70B-Chat` and `CodeLlama-34B-INST` by large margins.

| Model | Math Reasoning with Python | | | Question Answering with WikiSearch API | | MicroAvg |
| | M-GSM8K | M-MATH | M-TheoremQA | M-HotpotQA | M-MMLU | |
|---|---|---|---|---|---|---|
| Llama-2-70B-Chat | 33.33 | 3.00 | 2.04 | 27.91 | 42.11 | 20.25 |
| CodeLlama-34B-INST | 25.00 | 4.00 | 2.04 | 16.28 | 30.26 | 14.87 |
| Lemur-70B-Chat | **58.33** | **6.00** | **8.16** | **44.19** | **56.58** | **31.65** |
| gpt-3.5-turbo | 43.75 | 26.00 | 28.57 | 27.91 | 56.58 | 36.71 |
| gpt-4 | **93.75** | **57.00** | **57.14** | **46.51** | **80.26** | **66.77** |

## 4.3 SELF-DEBUG WITH ENVIRONMENT FEEDBACK

The technique of self-debug gains considerable traction in the realm of code generation (Olausson et al., 2023; Zhang et al., 2023). This method consists of models using feedback informa-

Table 5: Model performance with environment feedback. `Lemur-70B-Chat` demonstrates strong capabilities to comprehend environment feedback, achieving performance on par with `gpt-3.5-turbo`. *Due to overfitting to `[PYTHON]` tag, `CodeLlama-34B-INST` fails to follow feedback and generated parsable results (Wang et al., 2023b).

| Debug Environment | Python Interpreter | | Bash Terminal | Database | Robotics |
|---|---|---|---|---|---|
| Dataset | M-HumanEval | M-MBPP | IC-Bash | IC-SQL | RoboCodeGen |
| `Llama-2-70B-Chat` | 8.89* | 8.79* | 31.50 | 67.89 | 48.65 |
| `CodeLlama-34B-INST` | 2.22* | 2.20* | **36.00** | 67.79 | 64.86 |
| `Lemur-70B-Chat` | **46.67** | **17.58** | 34.50 | **73.79** | **75.68** |
| `gpt-3.5-turbo` | 37.78 | 25.27 | 46.51 | 72.82 | 83.78 |
| `gpt-4` | **73.33** | **52.75** | **48.52** | **84.41** | **83.78** |

tion, such as interpreter error tracebacks and database observations, to rectify any existing errors. This adaptive capacity is essential for agents because they have to constantly receive and react to feedback from the environment during interaction. As demonstrated in Table 5, the performance of the `Lemur-70B-Chat` significantly surpasses that of the `Llama-2-70B-Chat` and `CodeLlama-34B-INST` in interactive coding benchmarks. This underscores the importance of having balanced capabilities for interactive agents in such environments.

We further analyze the results of InterCode-SQL to understand how models incorporate environment feedback. In this setting, agents are provided with database schema and guidelines for SQL game interactions. Acting as agents, the models are tasked with querying databases and responding to questions in a multi-turn interaction environment. Figure 3 shows the Growth in Success Rate across models with each interaction turn. Lemur demonstrates robust performance in the first round, showing an initial performance that is comparable to that of `text-bison-001` and slightly lower than `gpt-3.5-turbo`. Nevertheless, Lemur improves consistently across ten interactive rounds, surpassing the performance of `gpt-3.5-turbo` eventu-

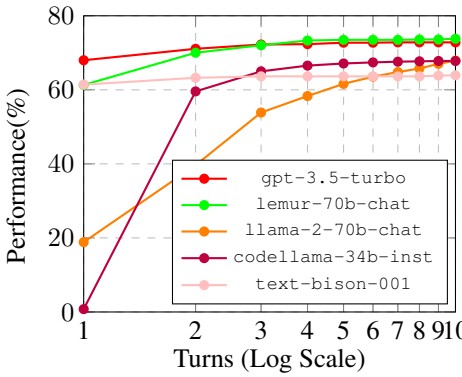

Figure 3: Success Rate with interaction turns across models in IC-SQL.

ally. In contrast, `text-bison-001`, which exhibits comparable initial performance with Lemur, does not show significant improvement. `Llama-2-70B-Chat`, while displaying consistent adaptability to feedback throughout the process, has a significant gap in initial performance due to inferior coding ability, hence its success rate remains relatively lower. `CodeLlama-34B-INST` hardly answers the questions correctly in the first round. We observe that this is because it blindly follows the advice in the game guide and stubbornly executes the `show tables` command first, instead of trying to understand the provided database structure. After an improvement in the second round, its performance returns to normal. However, the growth remained limited even after ten rounds of interaction, settling at par with `Llama-2-70B-Chat` and reflecting its relative weakness in adapting to environmental feedback. Additionally, we analyzed the problem difficulty influence in §D.2.

## 4.4 FOLLOW NATURAL LANGUAGE FEEDBACK

Following natural language feedback from users or other agents is an important ability for language agents. It requires agents to understand complex natural language instructions and convert them into symbolic executable sequences based on the current contexts. To evaluate the models' ability to follow natural language feedback, we follow the evaluation settings of MINT (Wang et al., 2023b), which measures language agents' ability to leverage natural language feedback using performance improvement. To provide natural language feedback in multi-turn settings, MINT uses GPT-4 to simulate a user providing helpful feedback on the solutions from evaluated language agents.

Table 7: Performance comparison of different models in partially observable environments InterCode-CTF, WebArena and ALFWorld.

| Model | Digital Env. | | Physical Env. |
|---|---|---|---|
| | IC-CTF | WebAreana | ALFWorld |
| Llama-2-70B-Chat | 9.00 | 1.72 | 21.64 |
| CodeLlama-34B-INST | 16.00 | 4.06 | 37.31 |
| Lemur-70B-Chat | **22.00** | **5.30** | **59.70** |
| gpt-3.5-turbo | 11.00 | 7.38 | 41.79 |
| gpt-4 | **37.00** | **10.59** | **84.33** |

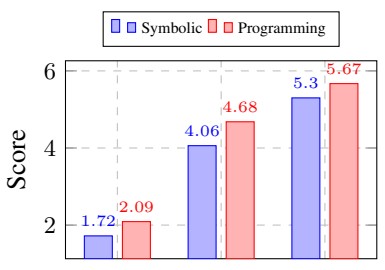

Figure 4: Performance Comparison of symbolic and programming representation for WebArena.

We evaluate models on MINT-Reasoning and MINT-Code with and without GPT-4 feedback and the experimental results are in Table 6. MINT-Reasoning includes five modified benchmarks: GSM8k, MATH, TheoremQA, HotpotQA, and MMLU. MINT-Coding includes modified HumanEval and MBPP. We find that all models can benefit from GPT-4, which means powerful GPT-4 as a teacher can provide helpful feedback even without ground truth. We also calculate $\Delta_{\text{feedback}}$, which indicates the absolute improvement thanks to GPT-4 feedback. According to the results from Table 6, Lemur-70B-Chat model obtain 8.19 in $\Delta_{\text{feedback}}$, which is significantly better than Llama-2-70B-Chat and CodeLlama-34B-INST.

Table 6: Comparative analysis of various models' Successful Rate (SR) on tasks related to Reasoning and Code Generation, with and without GPT-4 feedback. The table presents the performance metrics of each model under two conditions: 'no feedback' and 'with GPT-4 feedback'. The Micro Avg. column represents the average performance of the models across the tasks.

| Model | Feedback | Reasoning | Code Gen. | Micro Avg. | $\Delta_{\text{feedback}}$ |
|---|---|---|---|---|---|
| Llama-2-70B-Chat | no feedback | 20.25 | 8.82 | 16.81 | |
| | w/ GPT-4 feedback | 23.10 | 19.85 | 22.12 | 5.31 |
| CodeLlama-34B-INST | no feedback | 14.87 | 2.21 | 11.06 | |
| | w/ GPT-4 feedback | 20.25 | 3.68 | 15.27 | 4.20 |
| Lemur-70B-Chat | no feedback | 31.65 | 27.21 | 30.31 | |
| | w/ GPT-4 feedback | 35.76 | 44.86 | 38.50 | 8.19 |
| gpt-3.5-turbo | no feedback | 36.71 | 29.41 | 34.51 | |
| | w/ GPT-4 feedback | 50.32 | 38.97 | 46.90 | 12.39 |
| gpt-4 (upper-bound) | no feedback | 66.77 | 59.56 | 69.11 | N/A |

## 4.5 EXPLORE IN PARTIALLY OBSERVABLE ENVIRONMENTS

We evaluate language agents in varied environments and find that Lemur-70B-Chat exhibits balanced and commendable performance across all tested tasks. As shown in Table 7, it scored 22.00 in InterCode-CTF, showcasing its proficiency in multifaceted terminal skills and strategic adaptability. Moreover, we conducted further error analysis on CTF tasks in §D.3 and found that besides general skills of OS terminal, CTF tasks also require cybersecurity domain knowledge. This indicates the challenging requirements for agents to perform well in different environments. In WebArena, it achieved a score of 5.79, reflecting its adeptness in interpreting and executing advanced natural language commands in intricate web environments. In ALFWorld, it demonstrated superior planning and commonsense reasoning with a score of 59.70, successfully navigating and performing in simulated physical environments. While gpt-4 overall exhibits higher scores, Lemur-70B-Chat's consistent performance across diverse and partially observable environments underscores its versatility and potential in handling real-world, multifarious applications.

We also conduct experiments to explore different output formats for actions in the WebArena environment. Instead of merely prompting the language model to directly produce predefined actions

(e.g. type [id] [content] [press_enter_after]), we deterministically map the action space to Python representations (e.g. type(id:int, content:str, press_enter_after:bool)). We then prompt the language model to predict this representation and then parse the model's Python prediction into the predefined executable actions. Our findings, as shown in Figure 4 indicate that mapping to a Python representation leads to better performance than directly predicting actions. Such results suggest that carefully and rationally selecting intermediate representation, based on the pre-training corpus, can effectively enhance the model's performance as a language agent, aligning with prior findings (Hu et al., 2022).

## 5 RELATED WORK

**Transfer Learning on Code** The advancement in expansive language models (Devlin et al., 2019; Radford et al., 2019; Raffel et al., 2019; Brown et al., 2020) has catalyzed progress in transfer learning for code-related tasks, further enriched by the continuous pre-training paradigm (Gururangan et al., 2020). Hernandez et al. (2021) offered insights into the interplay between model size, training data, and performance in transferring language ability to code tasks. Several models have been introduced, exhibiting enhanced program synthesis and infilling/completion performance, by undergoing continual pre-training on extensive code data (Feng et al., 2020; Wang et al., 2021; Chen et al., 2021; Rozière et al., 2023). However, their intense focus on code often results in a compromise on natural language capabilities. Lemur addresses this by moderately augmenting a large model with a balanced mixture of code and natural language data, maintaining proficiency in both domains.

**Instruction Fine-tuning** The process of aligning LLMs to follow instructions, often referred to as instruction tuning, has been primarily directed towards NLP tasks (Wei et al., 2021; Wang et al., 2022b). Recent studies have sought to broaden the use cases of instruction tuning to involve a wider variety of general tasks (Ouyang et al., 2022). Self-instruct method generates instructions using seed instructions, aiding the understanding of how to adapt language models by fine-tuning them on instructions garnered from ChatGPT (Wang et al., 2022a; Zheng et al., 2023; Xu et al., 2023b; Mukherjee et al., 2023). We adopt a similar approach in tuning our model to follow instructions.

**Language Agents** Language agents are adept at following user instructions and engaging with environments to execute tasks. Recent trends in research and open-source communities have employed Large Language Models (LLMs) (Brown et al., 2020; Chen et al., 2021; Chowdhery et al., 2022; OpenAI, 2023) as the principal controllers for these agents (Yao et al., 2022b; Chase, 2022; Gravitas, 2023; Shinn et al., 2023; Wang et al., 2023a; Xu et al., 2023a; Lin et al., 2023; Yao et al., 2023). This is driven by the LLMs' demonstrated abilities in reasoning, planning, grounding, and code generation (Wei et al., 2022; Huang et al., 2022a; Ichter et al., 2022; Xie et al., 2023), crucial for comprehending user instructions, grasping the environmental context, and generating executable actions. Lemur seamlessly integrates capabilities in both text and code, enabling the generation of environment-grounded, executable actions essential for constructing language agents.

**Agent Evaluation Benchmarks** The rapid evolution of language agents requires an accurate and comprehensive assessment of agents. Recent works pose new dimensions that put language agents in web environments (Deng et al., 2023; Yao et al., 2022a; Zhou et al., 2023), interactive code environments (Yang et al., 2023), digital game (Fan et al., 2022), and household (Puig et al., 2018; Shridhar et al., 2020a;b) to finish certain task under natural language instruction. Apart from collecting new tasks, these agent benchmarks can be established by re-purposing and transforming existing datasets to agent datasets (Wang et al., 2023b; Yang et al., 2023; Liu et al., 2023). Our research combines these tasks, conducts extensive evaluation, and evaluates the model's capability to construct a language agent from multiple dimensions.

## 6 CONCLUSION

In conclusion, this research underscores the pivotal role of harmonizing natural and programming language proficiencies in the evolution of language models to sophisticated language agents. By developing Lemur and Lemur-Chat, we demonstrated that the meticulous amalgamation of these competencies allows for elevated performance in diverse environments and applications, narrowing the existent capability divide between open-source and proprietary models. We open-sourced both models, intending to foster further research in the field of language models for agents.

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

# A  LEMUR AND LEMUR-CHAT

## A.1  PRE-TRAINING

### A.1.1  PRE-TRAINING DATA

We performed pre-training on the top of Llama-2. The detailed statistics of pre-training data corpus is presented at below in Table 8.

Table 8: Distribution of data by Type and Source, as well as the sampling weights. Epochs are the number of passes over each constituent dataset during a full epoch over the data split.

| Type | Weights (%) | Source | Weights (%) | Effective Tokens (B) | Epoch |
|------|-------------|--------|-------------|----------------------|-------|
| Code | 90.9 | Python | 72.73 | 65.46 | 2.98 |
| | | SQL | 5.15 | 4.63 | 0.69 |
| | | Java | 1.82 | 1.64 | 0.06 |
| | | Shell | 1.82 | 1.64 | 1.26 |
| | | Notebook | 1.71 | 1.54 | 0.82 |
| | | JavaScript | 1.69 | 1.52 | 0.06 |
| | | C | 1.61 | 1.44 | 0.06 |
| | | PHP | 1.33 | 1.19 | 0.06 |
| | | CPP | 1.21 | 1.09 | 0.06 |
| | | Others | 1.73 | 1.55 | 0.06 |
| Text | 9.1 | RefinedWeb | 6.82 | 6.14 | – |
| | | Wikipedia | 0.64 | 0.57 | – |
| | | Books | 0.41 | 0.37 | – |
| | | ArXiv | 0.41 | 0.37 | – |
| | | StackExchange | 0.41 | 0.37 | – |
| | | DM Mathematics | 0.41 | 0.37 | – |
| Total | 100 | | 100.00 | 90.00 | |

### A.1.2  DATA MIXTURE RATIO DISCUSSION

The data mixture for continual pre-training is a complex trade-off between the model size, the vanilla performance of the model, data quality, learning efficiency, catastrophic forgetting etc Hernandez et al. (2021). To roughly estimate a good ratio of composing the training corpora, we conducted a continue pre-training study with Llama models on a set of different code-text ratios and found that the ratio of 10:1 is an efficient ratio for the Llama model to transfer from text to text-code balance.

Due to computational limits, it is difficult for us to conduct comprehensive experiments and perform systematic studies on this scale. However, we hope that our open-source effective settings and training checkpoints can benefit the community for continual exploration. We believe that a method to predict the optimal continue-pre-training data mixture ratio for a pair of domains to maximize their performance would be very meaningful and interesting, and it is still an open research question for the current large language model Hernandez et al. (2021); Chowdhery et al. (2022); Hoffmann et al. (2022); Aghajanyan et al. (2023).

When generalizing this approach to different models, the data mixture ratio and training steps need further adjustment. For example, smaller models have less capacity for harmonizing both text and code capabilities. Therefore, they may suffer from relatively obvious forgetting of natural language knowledge.. The strategy of data mixture for large language model pretraining is an open and valuable research question. We believe we will have a more efficient and predictable data mixture strategy in future studies.

### A.1.3  TRAINING DETAILS

We train our model on a TPUv4-512 pod. Our codebase is based on Jax and EasyLM (Geng, 2023). Following the pretraining methodology of Llama 2 (Touvron et al., 2023a), we used a batch size of

4M tokens. To improve training efficiency, we packed multiple shorter sequences into each batch entry when possible, an approach known as sequential packing (Raffel et al., 2019).

Optimization was performed with Adam using a peak learning rate of 4e-5 along with $\beta_1 = 0.9$ and $\beta_2 = 0.95$. Gradients were clipped at 1.0 to prevent exploding gradients. A cosine decay schedule was used for the learning rate, with a linear warmup of 2000 steps followed by decaying the learning rate to 10.% of its peak value at the end of training.

## A.2 INSTRUCTION FINE-TUNING

### A.2.1 INSTRUCTION FINE-TUNING DATA

**OpenORCA** The OpenOrca dataset is a collection of augmented FLAN Collection data. Currently 1M GPT-4 completions, and 3.2M GPT-3.5 completions. It is tabularized in alignment with the distributions presented in the ORCA paper (Lian et al., 2023) and currently represents a partial completion of the full intended dataset, with ongoing generation to expand its scope. The data is primarily used for training and evaluation in the field of natural language processing.

**OpenAssistant 1** OpenAssistant(OASST1) is a crowdsourced human-annotated assistant-style conversation corpus comprising 161,443 messages, enriched with 461,292 quality ratings. (Köpf et al., 2023). The data results from the collaboration of over 13,500 volunteers worldwide.

**ShareGPT and Chatlogs** We curated English human instructions from ShareGPT and Chatlogs for instruction-tuning. To filter English instructions, we utilized the langdetect package in Python and eliminated any instructions with non-English detection results. Additionally, considering that non-English instructions often contain consecutive non-English characters, we implemented a second filtering step, removing all sentences with three or more consecutive non-English characters. To ensure semantic diversity, we employed instructor(Su et al., 2023) to encode the filtered instructions, calculate cosine similarity, and remove instructions with a similarity score greater than 0.95. After deduplication, we obtained nearly 80K instances, with an average of about 6 rounds of high-quality data per instance.

**Evol-CodeAlpaca** We use two open-sourced Evolution-Instruct (Luo et al., 2023) datasets, i.e., Evol-Instruct-Code-80k-v1 and evol-codealpaca-v1, and an execution-verified Python dataset constructed by us. After applying the same deduplication method as the Text data, we obtained ~ 46K examples.

### A.2.2 INSTRUCTION FINE-TUNING DETAILS

We use Huggingface Transformers (Wolf et al., 2019) with Accelerate Library to fine-tune the Lemur model to obtain the Lemur-Chat model. We train on our data collection for two epochs. We use Adam optimizer with a learning rate of 2e-5 and a batch size of 128.

## B AGENT EVALUATION

**MINT** (Wang et al., 2023b) is a well-rounded evaluation that covers a range of tasks repurposed for multi-turn evaluation. It consists of three types of tasks, namely reasoning (MMLU (Hendrycks et al., 2021a), GSM8K (Cobbe et al., 2021), MATH (Hendrycks et al., 2021a), TheoremQA (Chen et al., 2023a), HotpotQA (Yang et al., 2018)), code generation (HumanEval (Chen et al., 2021), MBPP (Austin et al., 2021)), and decision-making (ALFWorld (Shridhar et al., 2020b)). To assess the proficiency of language models in employing tools, their reasoning process is reoriented to incorporate tool use. For instance, language models are prompted to utilize the Python calculator to work out mathematical problems, rather than supplying the answer outright. In code generation, the LLM is encouraged to incorporate Python interpreter messages to check generated code. To prevent any misunderstanding, we use the prefix "M-" to differentiate the original dataset from the MINT version in our paper.

**InterCode-Bash/SQL** (Yang et al., 2023) are two tasks that serve as an experimental platform that assesses the capacity of extensive language models to integrate feedback from the environment dur-

ing interactive coding. This benchmark evaluates models in a way of generating a series of actions under user instruction, and regards elements such as execution outcomes, and error backtracking, amongst others as environment observations.

**RoboCodeGen** (Liang et al., 2023) serves as a specialized evaluation framework focused on robotics-related tasks. It comprises three main types of questions that target spatial reasoning (e.g., identifying the closest point among a set of points), geometric reasoning (e.g., verifying if one bounding box is contained within another), and controls (e.g., PD control).

**InterCode-CTF** is a task in the InterCode evaluation suite. Capture the Flag (CTF) is a competitive cybersecurity game that requires LLMs to discover encrypted "flag" hidden within code snippets or file systems. Compared with Bash and SQL generation, CTF is much more complex, requiring agents to have knowledge of multiple coding languages, modularize higher-order objectives into sub-problems, create multi-step plans for solving each problem, and adjust strategies when a plan fails to provide any useful insights.

**WebArena** (Zhou et al., 2023) creates self-hostable websites of four popular categories by simulating real-world equivalent functionalities and data. To simulate human problem-solving abilities, WebArena also embeds tools and knowledge resources as standalone websites.

**ALFWorld** (Shridhar et al., 2020b) is a synthetic environment benchmark adapted from Alfred (Shridhar et al., 2020a) in the text-based interface where agents need to navigate in simulated households (e.g., go to coffee table 1, pick up paper 2, use desk lamp 1) and achieve high-level goals (e.g., check the paper under the desk lamp). Task instances may involve over 50 locations and 50 steps to solve, thus challenging the agent's ability to plan, track sub-goals, and conduct systematic exploration.

## C  BASELINE MODELS

`StarCoder-15B` Li et al. (2023) is a model with 15.5B parameters and 8k context length. It is first trained on 1 trillion tokens sourced from The Stack Kocetkov et al. (2022), a large collection of permissively licensed GitHub repositories, and then finetuned on 35B Python tokens.

`StarCoderPlus-15B` Li et al. (2023) is similar to StarCoder-15B, except that it is finetuned in RefinedWeb Penedo et al. (2023), The Stack Kocetkov et al. (2022) and Wikipedia dataset. It is expected to have more balanced text and code capabilities.

**Llama-2** Touvron et al. (2023b) is a model with 70B parameters trained on 2 trillion tokens from public sources. Its training corpus is mostly natural language texts, which enables its strong abilities in textual understanding.

`Llama-2-70B-Chat` Touvron et al. (2023b) finetunes Llama-2-70B and is optimized for dialogue use cases.

`CodeLlama-34B` Roziere et al. (2023) finetunes Llama-2-34B on code-intensive corpus with 500B tokens.

`CodeLlama-34B-INST` Roziere et al. (2023), Code Llama - Instruct is based on Code Llama and fine-tuned with an additional approx. 5B tokens to better follow human instructions.

`WizardCoder-15B` Luo et al. (2023) was trained with instrucions on code-intensive datasets. It achieved superior performance on popular code generation benchmarks including HumanEval, HumanEval+, MBPP, and DS1000.

## D  ERROR ANALYSIS ON LANGUAGE AGENT TASKS

### D.1  AUGMENT WITH TOOLS: A CASE STUDY ON MINT-MATH

In Table 9, we report the percentage of examples in MINT-MATH dataset with execution errors and invalid actions. From the results, we can see that `Llama-2-70B-Chat` produces the most execution error, followed by `CodeLlama-34B-INST`, `Lemur-70B-Chat`, `gpt-3.5-turbo` and `gpt-4` has the fewest error. This is aligned with the performance of task success rates reported

Table 9: Model error rate in MATH evaluation set by causes

| Model | Execution error | Invalid action |
|-------|-----------------|----------------|
| Llama-2-70B-Chat | 40.00 | 8.00 |
| CodeLlama-34B-INST | 38.00 | 33.00 |
| Lemur-70B-Chat | 29.00 | 26.00 |
| gpt-3.5-turbo | 17.00 | 15.00 |
| gpt-4 | 7.00 | 2.00 |

in Table 4, which indicates that execution error is an important factor in explaining the performance difference between models.

On the other hand, the rate of invalid actions among different models also follows similar trends, except that Llama-2-70B-Chat achieves extremely low error rate. With further investigation into the example output, we find out that Llama-2-70B-Chat usually writes trivial and ineffective actions, which does not contribute to solving the problem.

### D.2 SELF-DEBUG WITH ENVIRONMENT FEEDBACK: A CASE STUDY ON INTERCODE-SQL

Table 10: Model accuracy in different spectrum of Spider evaluation set

| Model | Easy | Medium | Hard | Extra | All |
|-------|------|--------|------|-------|-----|
| Llama-2-70B-Chat | 89.92 | 69.73 | 59.77 | 38.55 | 67.89 |
| CodeLlama-34B-INST | 90.73 | 67.49 | 63.79 | 38.55 | 67.80 |
| Lemur-70B-Chat | 92.74 | 74.89 | 70.69 | 45.78 | 73.79 |
| gpt-3.5-turbo | 92.74 | 74.89 | 67.24 | 43.37 | 72.82 |
| gpt-4 | 95.16 | 82.96 | 86.21 | 68.67 | 84.14 |

In Table 10, We report detailed scores of different models in various spectrums of the Spider evaluation set divided by difficulty level. According to Table 10, as the difficulty level increases, the performance gap between gpt-4 and Lemur-Chat gradually widens. When evaluated with easy questions, the model usually solves the problem in the first round; while in the difficult split, the model needs to iteratively incorporate the environment feedback. The larger gap between Lemur-Chat and gpt-4 indicates better abilities of gpt-4 to perform multi-turn interaction with the environment.

### D.3 EXPLORE IN PARTIALLY OBSERVABLE ENVIRONMENT : A CASE STUDY ON CTF

We manually researched the CTF (Capture The Flag) tasks, which is a popular competition program originating from cybersecurity. We manually labeled the problems in 100 CTF tasks and divided each problem into 6 categories. Figure 5 shows the performance comparison between Lemur-70B-Chat and gpt-4. We found that gpt-4 and Lemur-70B-Chat perform significantly better in the "general skills" category than in domain-specific fields such as "cryptography" and "reverse engineering". This means current language agents methods easily fail on domain-specific scenarios, which guide the future research of language agents.

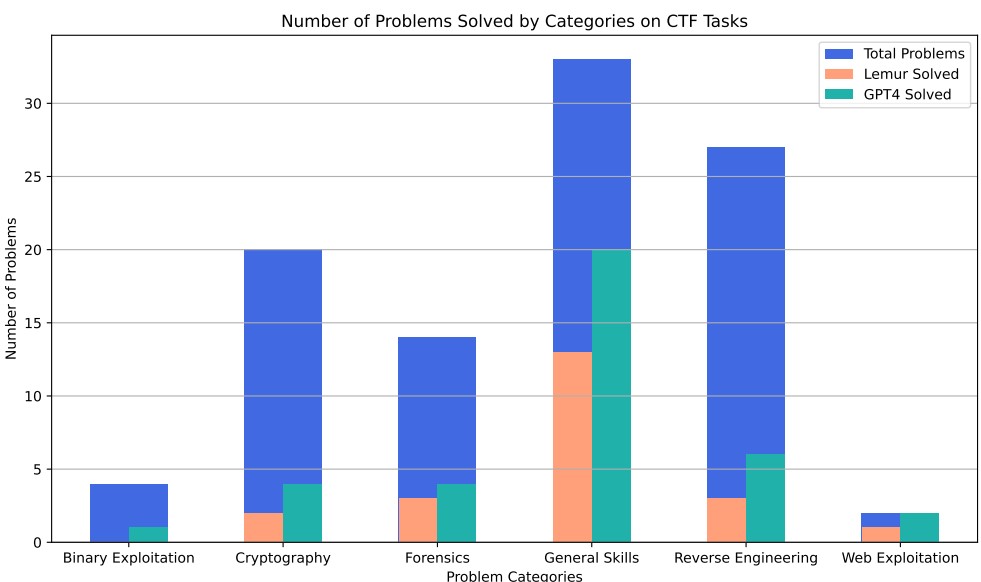

Figure 5: The number of problems solved by `Lemur-70B-Chat` and `gpt-4`. The blue part indicates the total number of problems.

