# OpenReview forum: "Lemur: Harmonizing Natural Language and Code for Language Agents"
_ICLR.cc/2024/Conference — ICLR 2024 spotlight_

### Official Review · Reviewer_QCzC · 2023-11-01

**Soundness:** 4 excellent
**Presentation:** 4 excellent
**Contribution:** 4 excellent
**Rating:** 8
**Confidence:** 4

**Summary:**

The paper introduces Lemur and Lemur-Chat, openly-accessible large language models that harmonize natural language with code capabilities. The Lemur models are trained on the basis of Llama-2, with a code-centric pre-training stage with a code-to-text ratio of 10:1 for code-text harmonization, and a supervised instruction fine-tuning stage. The authors conduct systematic and comprehensive evaluations of Lemur models on diverse benchmarks, consisting of fundamental code/language benchmarks and pratical scenarios that connect LLMs to environments. The paper categorizes the capabilities of LLM agents in four aspects: agument with tools, self-debug, following feedback, and exploring environments. Over extensive benchmarks, the experimental results demonstrate harmonized capabilties between natural language and codes, and show that the Lemur models consistently outperform their counterparts on a wide range of tasks.

**Strengths:**

- The idea of harmonizing the natural language and coding capabilities of LLMs is nice. With carefully designed code-to-text ratio and the selection of training data, the resulting Lemur models achieve a harmonious blend of language and coding capabilities.
- The resulting Lemur models achieve competitive performance on language-coding tasks against gpt-3.5-turbo. The open-sourced Lemur models will be useful for the research community, and would be foundation models to develop agents.
- The experiments are solid and evaluations are systematically organized. The Lemur models are evaluated in a clear and comprehensive evaluation process. The evaluation consists of the evaluations in each domain of code or language, and diverse code-language tasks that are grouped into 4 types of skills, establishing a good evaluation procedure for language-code LLM agents.
- The paper is clear and concise with well-structured evaluations.

**Weaknesses:**

- As mentioned in Introduction, the paper has offered valuable insights on synergy, but it is unclear what the insights exactly are. I would suggest clearly presenting the insights instead of letting readers find where is the insights across the paper.
- Minor: In Figure 2, the capitalizations are not consistent. (Use->,  run->); Section 4.5: mapp -> map; Section 4.5: intermm.

**Questions:**

- Why is a large proportion of the pre-training data is in Python?
- Is the harmonization controlled by the text-to-code ratio? How did you come up with the idea of setting a ratio of 10:1?

---

> ### Author Response · Authors · 2023-11-17
> **Response to reviewer QCzC**
>
> Thank you for your reviews and comments to the paper! We are glad to hear that you appreciate the idea of harmonizing natural language and find our results solid and competitive. Below we address the concerns and questions raised in the review.
>
> ---
>
> **Weakness 1:** *Lack of Clear Presentation of Insights*
>
> **Response:** We appreciate the detailed feedback you provided on our writing and clarity. Based on these suggestions, we have highlighted the most important insights about harmony ability with specific descriptions in the introduction. Additionally, we have added error analysis as well as additional findings and insights.
>
> ---
>
> **Weakness 2:** Minor Typos in the Paper
>
> **Response:** We have corrected the typos in our revised version. Thanks a lot for pointing them out!
>
> ---
>
> **Question 1:** Reason for High Proportion of Python in Pre-training Data
>
> **Question 2:** Harmonization and the Text-to-Code Ratio Decision
>
> **Response:** We thank you for your valuable questions on our technical details!
>
> - The data mixture strategy is a complex trade-off between the model size, the vanilla performance of the model, data quality, learning efficiency, catastrophic forgetting etc.
> - To roughly estimate a good code-to-text ratio for composing the training corpora, we conducted a continue pre-training study with Llama on a set of different ratios and found that the ratio of 10:1 is an efficient ratio for the Llama model to transfer from text to text-code balance, which aligns with the findings of recent work like CodeLlama.
> - Among all programming languages, we focus on scripting or interpreted languages (Python, SQL, Bash, Perl, etc.) because agent models are often executed in interactive scenarios. Unlike compiled languages like C++, the interpreted nature of scripting languages allows immediate execution and easy modification, which is essential for dynamic interaction in language agent scenarios.
> - Due to computational limits, it is difficult for us to conduct comprehensive experiments and perform systematic studies on this scale. However, we hope that our open-sourced effective settings and training checkpoints can benefit the community for continual exploration.
> - We believe that a method to predict the optimal continue-pre-training data mixture ratio for a pair of domains to maximize their performance would be very meaningful and interesting, and it is still an open research question for the current large language model.
>
> We have added this explanation in the appendix of our revised version. Thanks for your valuable question!

---

> > ### Comment · Reviewer_QCzC · 2023-11-22
> >
> > Thank you for addressing my comments.

---

### Official Review · Reviewer_JHRb · 2023-11-02

**Soundness:** 3 good
**Presentation:** 3 good
**Contribution:** 2 fair
**Rating:** 6
**Confidence:** 3

**Summary:**

This paper proposes Lemur and Lemur-Chat language models, emphasizing their combined proficiency in both natural language understanding and coding capabilities. These models are designed to bridge the gap between understanding human interactions and manipulating code, aiming to serve as versatile language agents.

what contributions does it make:
1.The proposed models Lemur and Lemur-Chat narrow the gap with proprietary models in terms of agent abilities, leveraging its harmonization of both natural language and programming languages.
2.Provide comprehensive evaluations of language and coding abilities.
3.These models are open-source, providing a valuable resource for the community and potentially contributing to the development of advanced open-source agents that can be seamlessly reasoned, planned, and run in a variety of environments.

**Strengths:**

1.It improves the coding ability while maintaining the reasoning ability of Llama-2.
2.The Lemur is pre-trained and fine-tuned using a rich dataset that includes text and code, ensuring a balance of performance across a variety of text and coding benchmarks.
3.The model showcases proficiency in agent tasks, encompassing human communication, tool usage, and interaction across observable environments.

**Weaknesses:**

1.It seems that pre-training takes the responsibility to gain the coding ability, and the supervised fine-tuning takes the responsibility to gain the natural language ability, while it is vague how the proposed model balance these two abilities.
2.As shown in Tables 4, 5, and 7, the performance of the proposed model Lemur-70B-Chat falls short when compared to GPT-4 and this discrepancy in performance lacks an explanatory or discussion.
3.Table 3 lists three baseline models—StarCoder-15B, StarCoderPlus-15B, and WizardCoder-15B—without corresponding explanations or references in the provided context.
4.Table7 does not have references and analysis.

**Questions:**

1.GPT4 also integrates text and code capabilities, what are the advantages of this paper?

---

> ### Author Response · Authors · 2023-11-17
> **Response to reviewer JHRb (1/2)**
>
> Thank you for your insightful reviews and comments. We are glad to hear that you found our evaluations comprehensive and considered Lemur as a valuable resource for developing advanced open-source agents. Below we address the concerns and questions raised in the review.
>
> ---
>
> **Weakness 1:** How the proposed training procedures balance these two abilities
>
> **Response:**
> - The strategy we use to balance the two abilities involves two-stage training. First, we train a good text model LLaMA-2-70B on code-intensive tasks, which aims to enhance its coding abilities. This gives Lemur-70B. Second, we finetune Lemur-70B to improve its instruction-following capabilities on both text and code scenarios.
> - From the results we present in Table 3 (which we also copy here for easier reference), it can be seen that Lemur-70B outperforms LLaMA-2-70B in code tasks, but maintains a good balance in text tasks. Additionally, thanks to its instruction-following capability, Lemur-70B-Chat achieves better performance in all scenarios.
>
>
> | Model | Text |  |  | Code |  |  |  |  | Average |
> | --- | --- | --- | --- | --- | --- | --- | --- | --- | --- |
> |  | QA | Reason | Math | Python |  | SQL | MCode | DS |  |
> |  | MMLU | BBH | GSM8K | HumanEval | MBPP | Spider | MultiPL-E | DS-1000 |  |
> | LLaMA-2-70B | 68.9 | 51.2 | 56.8 | 30.5 | 45.4 | 60.0 | 24.4 | 11.3 | 43.6 |
> | Lemur-70B | 64.5 | 51.6 | 54.9 | 35.4 | 53.2 | 62.8 | 30.4 | 30.7 | 47.9 |
> | Lemur-70B-Chat | 65.3 | 61.9 | 66.3 | 61.0 | 55.5 | 62.5 | 42.9 | 34.5 | 55.0 |
>
> ---
>
> **Weakness 2:** Explanation or discussion of the performance differences with GPT-4
>
> **Response:** We appreciate your attention to the differences between GPT-4 and our Lemur model! To better address your concern about performance gaps and guide the development of future language agent models, we added a 3-part human error analysis section in Appendix D in the revised paper. We hope this section can provide a detailed analysis of the three scenarios you mentioned, and also reflect the underlying reasons for the discrepancy between GPT-4 and our model. We summarize them as the following:
> - **Tool Usage ability:** We found that very complex and lengthy problems make it difficult for the models except GPT-4 to generate executable agent actions in the correct format. We found this problem when studying the M-MATH results. The long problems from MATH include solving equations described using LaTeX syntax, which is long and noisy. This misleads language models, which often generate unparsable responses or wrong actions. We hope to keep improving the ability of instruction following abilities in these scenarios. Here is part of our newly added Table 9:
>
> | Model                        | Execution error | Invalid action |
> |------------------------------|-----------------|----------------|
> | Lemur-70B-Chat             | 29.00           | 26.00          |
> | gpt-3.5-turbo                     | 17.00           | 15.00          |
> | gpt-4                      | 7.00            | 2.00           |
>
> - **Self-debugging ability:** Although the Lemur model is close to the performance of GPT-3.5-turbo in many environments of this scenario, we found that there is still a significant gap between Lemur and GPT-4 in challenging problems. We take InterCode-SQL as an example to study this issue. We categorize the problems of the SQL problems into different levels. According to the table, as the difficulty level increases, the performance gap between GPT-4 and Lemur gradually widens. This guides us to pay more attention to challenging cases. Here is part of our newly added Table 10:
>
> | Model                        | Easy  | Medium | Hard  | Extra | All   |
> |------------------------------|-------|--------|-------|-------|-------|
> | Lemur-70B-Chat             | 92.74 | 74.89  | 70.69 | 45.78 | 73.79 |
> | gpt-3.5-turbo                     | 92.74 | 74.89  | 67.24 | 43.37 | 72.82 |
> | gpt-4                      | 95.16 | 82.96  | 86.21 | 68.67 | 84.14 |
>
>
> - **Exploring in Partially-observable environment:** We manually researched the CTF (Capture The Flag) tasks, which is a popular competition program originating from cybersecurity. We manually labeled the problems in 100 CTF tasks and divided each problem into 6 categories. We found that GPT-4 and Lemur perform significantly better in the "general skills" category than in domain-specific fields such as "cryptography" and "reverse engineering". This means current language agents methods easily fail in domain-specific scenarios, which guide the future research of language agents. Please refer to our newly added Figure 5.
>
> Please refer to Appendix D of our revised paper for more detailed information. We hope the added concrete detailed analysis can address your concerns!

---

> > ### Author Response · Authors · 2023-11-17
> > **Response to reviewer JHRb (2/2)**
> >
> > **Weakness 3&4:** Table 3 Lack of Explanations for Baseline Models; Missing References and Analysis in Table 7
> >
> > **Response:**
> > - We added references and explanations to baseline models StarCoder-15B, StarCoderPlus-15B, and WizardCoder-15B in Section 4.1 and Appendix C.
> > - In Section 4.5, we analyze the results shown in Table 7. In Appendix B, you may find the reference and introduction to the tasks InterCoder-CTF, WebArena, and ALFWorld. We also supplement the descriptions of baseline models in Appendix C.
> >
> > ---
> >
> > **Question 1:** Advantages over GPT-4's Integration of Text and Code Capabilities
> >
> > **Response:** Thanks for your comments! We would like to discuss our value in two aspects: a research study on language agents with insights and open-sourced models.
> >
> > - **Comprehensive language agent research study with insights:** Previous works usually studied language agents with specific tools and environments. In our work, we establish a comprehensive language agent study by organizing diverse environments to evaluate key agent skills in different scenarios. We are glad that this is recognized by all the reviewers.
> > - The insight into harmonious abilities motivated us to develop Lemur models, and our experiments indicate its direct importance to language agents compared with unbalanced models like Llama or CodeLlama. To provide more insights into language agent development,  we have added more human analysis in the appendix of this revised version, indicating the direction to further develop open language agents.
> > - **Open-sourcing effort value for community:** Current AI agent research largely relies on prompting close-sourced models like GPT-4, GPT-3.5 Turbo, and Claude-2. There are a lot of limitations when accessing close-sourced models by APIs, such as high cost and difficulties in custom training. We made expansive efforts to develop and open-source Lemur models. We believe that it will help the AI agent community develop and explore LLM-based agents by not only prompting models but also conducting agent training with different learning methods like RL. We are grateful to the Reviewer `QCzC` who also mentioned this point.
> > - In addition, we enjoy the natural advantages of open-source models, such as local deployment for privacy protection, transparent construction pipeline, and open-source community collaboration.

---

### Official Review · Reviewer_nTpx · 2023-11-03

**Soundness:** 2 fair
**Presentation:** 3 good
**Contribution:** 2 fair
**Rating:** 6
**Confidence:** 4

**Summary:**

This paper presents Lemur and Lemur-Chat, large language models that exhibit balanced proficiency in both language and coding. The paper further trains LLAMA on a corpus with a code-to-text ratio of 10:1 and fine-tunes the model on four instruction datasets. The paper evaluates these two models across a broad spectrum of tasks, which includes text benchmarks (such as MMLU, BBH, etc.) and code benchmarks (such as HumanEval, MBPP, MultiPL-E, etc.). Moreover, the paper demonstrates that these models perform exceptionally well in language agent scenarios, such as augmenting with tools, self-debugging with environment feedback, adhering to natural language feedback, and exploring in partially observable environments.

**Strengths:**

This article validates the effectiveness of the Lemur model on a large number of benchmarks and verifies the importance of balanced language and coding capabilities for language agent scenarios.

**Weaknesses:**

1) The technical contribution of this article is quite limited, it merely continues training the LLAMA model on a mixture of text and code data and instruction tuning on four datasets.
2) When comparing performance on the code benchmark, the authors use a large 70B model, but the code-specific models they compare with are mostly 15-30B in size, which makes the comparison somewhat unequal.

**Questions:**

Why you use 10:1 text-to-code ratio in your pretraining data?

---

> ### Author Response · Authors · 2023-11-17
> **Response to reviewer nTpx**
>
> Thanks for your time in reviewing and providing feedback for our work! We greatly appreciate your recognition of the effectiveness of the Lemur model and its balance in language agent scenarios. We have also noted your concerns about our technical contribution and comparison strategies in our work. We hope to address these concerns below:
>
>
> **Weakness 1:** *Limited technical contribution*
>
> **Response:** Besides the technical contribution of demonstrating our training procedure, We would also like to highlight our research value and community benefits, including research study and insights to guide future language agent development, establishing comprehensive language agent evaluation, and open-sourcing models with superior performance for the language agent community. We are very grateful to `Ycpy`, `JHRb`, and `QCzC` for mentioning these contributions.
>
> - **Research insights on language agent development:** Our insights into harmonious abilities motivated us to develop Lemur models, and our experiments indicate its direct importance to language agents compared to unbalanced models like Llama or CodeLlama. To provide more insights into language agent development,  We have further added more human analysis in the appendix of the revised version, indicating the direction to further develop open language agents.
> - **Comprehensive language agent research study:** Previous works usually studied language agents with specific tools and environments. In our work, we establish a comprehensive language agent study by organizing diverse environments to evaluate key agent skills in different scenarios. We are glad that this is recognized by all reviewers.
> - **Open-sourcing effort for language agents community as a key technical contribution:** Current AI agent research largely relies on prompting close-sourced models like GPT-4, GPT-3.5 Turbo, and Claude-2. There are a lot of limitations when accessing close-sourced models by APIs, such as high costs and difficulties in custom training. We made expansive efforts to develop and open-source Lemur models. We believe that it will help the AI agent community to develop and explore LLM-based agents by not only prompting models but also conducting agent training with different learning methods like RL. By being openly available, Lemur models can be accessed by individuals and organizations that might not have the resources to develop their own models from scratch, democratizing access to agent models.
>
> ---
>
> **Weakness 2:** *Unequal Model Size Comparison on the code benchmark*
>
> **Response:** Thanks for your detailed feedback on the model comparison!
> - To better track the performance of our model on code benchmark, we selected CodeLlama-34B and CodeLlama-34b-Instruct as representative baselines, which were the largest and most capable open-sourced code models.
> - We would also like to highlight that our motivation for training the Lemur models is to harmonize the model’s natural language and code capabilities for agent tasks. Being the most capable model on code benchmarks may not be our focus. Instead, we may emphasize the model’s overall performance in both natural language, code, and agent tasks.
>
> ---
>
> **Question 1:** *Reason for 10:1 Code-to-Text Ratio in Pretraining*
>
> **Response:**
> - The ratio of code to text is a complex trade-off between the model size, the vanilla performance of the model, data quality, learning efficiency, catastrophic forgetting, etc.
> - To roughly estimate a good ratio of composing the training corpora, we conducted a continue pre-training study with Llama models on a set of different code-text ratios and found that the ratio of 10:1 is an efficient ratio for the Llama model to transfer from text to text-code balance.
> - Due to computational limits, it is difficult for us to conduct comprehensive experiments and perform systematic studies on this scale. However, we hope that our open-sourced effective settings and training checkpoints can benefit the community for continual exploration.
> - We believe that a method to predict the optimal continue-pre-training data mixture ratio for a pair of domains to maximize their performance would be very meaningful and interesting, and it is still an open research question for the current large language model.
>
> We have added this explanation in Appendix A of our revised version. Thanks for your valuable question!

---

> ### Comment · Reviewer_nTpx · 2023-11-20
>
> Thanks for your response. I will increase the score.

---

### Official Review · Reviewer_Ycpy · 2023-11-04

**Soundness:** 4 excellent
**Presentation:** 4 excellent
**Contribution:** 4 excellent
**Rating:** 8
**Confidence:** 3

**Summary:**

This paper proposes two models Lemur and Lemur-Chat by training on a combined data of natural language and programming languages. Comprehensive experiments show that the proposed models show superior performance on 12 agent benchmarks.

**Strengths:**

**Originality:** This paper proposes a novel way of training LLMs with code + text data to design language agents.

**Quality:** There are detailed studies included in the paper about how training LLMs can be beneficial to solve both the language and agent tasks.

**Clarity:** The paper is well-written and easy to follow.

**Weaknesses:**

**Ambiguous Motivation:** I am fully not convinced with the sentence "for the construction of language agents, it is imperative for language models to possess harmonized capabilities in both natural language and programming languages." Its unclear how programming languages correlate with language understanding. In fact in the context of linguistics (morphology, syntax and semantics), programming languages might not satisfy any of them. I believe the authors should provide more context for it. Although the experimental results show that Lemur-Chat outperforms on majority of the datasets, correlation does not imply causation.

**Questions:**

1. Is there any reason to choosing scripting languages?
2. Is the performance replicable for base models other than Llama?
3. How much does the size of Llama matter for experiments? Can the same pipeline be replicated for smaller Llama versions?

---

> ### Author Response · Authors · 2023-11-17
> **Response to reviewer Ycpy**
>
> Thank you for recognizing our work! We are delighted to hear your appreciation for our training methods in designing and building language agents, as well as the recognition of our detailed study on language agent abilities. At the same time, we are happy to further explain the following topics regarding our motivation and the questions you have:
>
> ---
>
> **Weakness 1:** *Ambiguous Motivation for Integrating Programming Language Skills in Language Agents*
>
> **Response:** Thank you for your feedback regarding the motivation in our paper! We would like to further clarify our motivation:
> - The most significant difference between agents and chatbots is that agents not only need to communicate with human users using natural language, but also need to interact with complex environments using executable programming languages. For example, the agent of web browsing accepts user instructions and controls the browser using code to complete tasks in multiple steps.
> - Therefore, this motivates us to directly enhance the code capabilities to help agents manipulate the environment via code. We would like to show the direct importance of programming language in agent scenarios, rather than the relevance or benefits of programming languages to natural languages.
>
> ---
>
> **Question 1:** *Choice of Scripting Languages for Language Agents*
>
> **Response:**
> - In section 2.1, we briefly mentioned that, among all languages, we focus on scripting or interpreted languages (Python, SQL, Bash, Perl, etc.) because agent models are often executed in interactive scenarios. Unlike compiled languages like C++, the interpreted nature of scripting languages allows immediate execution and easy modification, which is essential for dynamic interaction in language agent scenarios.
> - Another interesting finding is that, although the training corpora of Lemur contain a large proportion of scripting languages, we observed a general performance improvement on mainstream programming languages compared to the original LLaMa-2. The following are the detailed results of MultiPL-E in Table 3, which is a multi-programming language evaluation benchmark.
>
> | Model | C++ | Java | PHP | TypeScript | C# | Bash | Average |
> | --- | --- | --- | --- | --- | --- | --- | --- |
> | LLaMA-2-70B | 30.40 | 31.70 | 34.20 | 15.10 | 25.90 | 8.90 | 24.4 |
> | Lemur-70b-v1 | 34.78 | 35.40 | 35.40 | 35.40 | 25.47 | 15.82 | 30.4 |
>
> ---
>
> **Question 2&3:** *Replicability for those models other than Llama or smaller than 70B.*
>
> **Response:**
> - Harmonizing model abilities by data mixture is a general and effective approach. For those base models other than Llama or smaller than 70B, we believe that it is generally capable of harmonizing natural language and code capabilities.
> - When generalizing this approach to different models, the data mixture ratio and training steps need further adjustment. For example, smaller models have less capacity for harmonizing both text and code capabilities. Therefore, they may suffer from relatively obvious forgetting of natural language knowledge. The strategy of data mixture for large language model pretraining is an open and valuable research question. We believe we will have a more efficient and predictable data mixture strategy in future studies.
>
> ---
>
> We hope our explanation could address your concern! We further revised our paper and appendix to clarify our motivation, incorporating your valuable inputs to better articulate our rationale.

---

### Author Response · Authors · 2023-11-17
**Updated Manuscript and Response to All Reviewers**

We thank all the reviewers for their feedback and constructive comments!  We are pleased to hear the recognition from reviewers. We appreciate `Ycpy` and `JHRb` acknowledging our innovative training approach; `nTpx`, `JHRb`, and `QCzC` recognizing the models' balanced proficiency in language and coding; `Ycpy` and `QCzC` providing positive feedback on the comprehensive evaluation and superior performance; `nTpx`, `QCzC` noting the effectiveness in language agent scenarios and competitiveness against models like GPT-3.5-turbo; and `JHRb` and `QCzC` highlighting the contribution of open-source to the community. We sincerely thank all the reviewers for their insightful comments and constructive feedback.

We have also noticed the concerns and questions of each reviewer, and we have responded to each question separately. In addition, based on the valuable feedback, we have revised our paper including the following improvements (updates are shown in purple for clarity):
- Updated Introduction to present our insights more clearly.
- Updated Section 2.1 and Added Appendix A.1 to include more explanations about the data mixture and the focus of scripting languages.
- Updated Section 4.1 to include more explanations and analysis of results in Table 3. Fixed the reference of Table 7.
- Added appendix C to include more descriptions and references of baseline models.
- Added Appendix D and updated Section 4 to include manual error analysis and provide more insights for future research.

We would like to thank once again all the reviewers for their time and efforts in helping us improve the paper! If you need any further explanations, please do not hesitate to let us know.

---

### Meta-Review · Area_Chair_fCVj · 2023-12-06

**Metareview:**

Summary:

This paper proposes Lemur and Lemur-Chat, large-scale language models tailored to harmonize capabilities in natural language understanding and programming. The Lemur models are trained on the basis of Llama-2, with a code-centric pre-training stage with a code-to-text ratio of 10:1 for code-text harmonization, and a supervised instruction fine-tuning stage, which is then demonstrated through superior performance on a variety of agent benchmarks. The paper is generally well-written and organized and authors put forward comprehensive experiments to substantiate their claims.

Strengths:

1)The paper introduces a fresh approach to training language models with a combined dataset of natural language and programming language, which is an innovative step forward in the field of language agents.

2)The empirical studies outlined in the paper are meticulous and provide substantive evidence for the efficacy of the proposed models. The extensive evaluations across 12 agent benchmarks highlight the models' balanced capabilities.

3) Lemur and Lemur-Chat are open-sourced, offering a significant resource to the research community and enabling further exploration, development, and transparency in the field of AI agents.


Weaknesses:

1. Initial concerns centered around how programming language integration is essential for language understanding within agent contexts. While the authors addressed this in their response, it is critical that such a central motivation be clear in the paper itself.

2. Some reviewers perceive the technical contribution as incremental since it extends an existing model (LLaMA) with further training on mixed data. The authors’ response clarified the broader implications and contributions; nevertheless, the paper would benefit from a more pronounced delineation of the novelty beyond technical experimentation.

3. Performance Analysis vs. GPT-4: The paper initially lacked a detailed analysis of its models' performance against leading models like GPT-4. While subsequent author responses have added more context, including such analysis in the main content of the paper is important to frame the work in the competitive landscape adequately.

**Justification For Why Not Higher Score:**

There is still a significant gap between Lemur and GPT-4 in challenging problems

**Justification For Why Not Lower Score:**

All reviewers voted to accept this paper, and the reviewers' comments were well addressed.  This paper shares good insight for research insights on language agent development and the comprehensive language agent research study is also very valuable.

---

### Decision · Program_Chairs · 2024-01-16

Accept (spotlight)